# Association between Primary Care Utilization and Emergency Room or Hospital Inpatient Services Utilization among the Middle-Aged and Elderly in a Self-Referral System: Evidence from the China Health and Retirement Longitudinal Study 2011–2018

**DOI:** 10.3390/ijerph191912979

**Published:** 2022-10-10

**Authors:** Siman Yang, Mengping Zhou, Jingyi Liao, Xinxin Ding, Nan Hu, Li Kuang

**Affiliations:** 1Department of Health Administration, School of Public Health, Sun Yat-Sen University, Guangzhou 510080, China; 2Department of Medical Epidemiology and Biostatistics, Karolinska Institute, 17177 Stockholm, Sweden; 3Department of Biostatistics, FIU Robert Stempel College of Public Health and Social Work, Miami, FL 33199, USA; 4Department of Family and Preventive Medicine and Population Health Sciences, University of Utah School of Medicine, Salt Lake City, UT 84132, USA

**Keywords:** primary care, healthcare service utilization, middle-aged and elderly individuals, CHARLS

## Abstract

With rapid economic growth and aging, hospital inpatient and emergency services utilization has grown rapidly, and has emphasized an urgent requirement to adjust and optimize the structure of health service utilization. Studies have shown that primary care is an effective way to reduce inpatient and emergency room (ER) service utilization. This study aims to examine whether middle-aged and elderly individuals who selected primary care outpatient services in the last month had less ER and hospital inpatient service utilization than those who selected hospitals outpatient services via the self-referral system. Data were obtained from four waves of the nationally representative China Health and Retirement Longitudinal Study (CHARLS). We pooled respondents who had outpatient visits and were aged 45 years and above. We used logistic regressions to explore the association between types of outpatient and ER visits or hospitalization, and then used zero-truncated negative binomial regression to examine the impact of outpatient visit types on the number of hospitalizations and the length of hospitalization days. A trend test was used to explore the trend of outpatient visit types and the ER or hospital inpatient services utilization with the increase in outpatient visits. Among the 7544 respondents in CHARLS, those with primary care outpatient visits were less likely to have ER visits (adjusted OR = 0.141, 95% CI: 0.101–0.194), hospitalization (adjusted OR = 0.623, 95% CI: 0.546–0.711), and had fewer hospitalization days (adjusted IRR = 0.886, 95% CI: 0.81–0.969). The trend test showed that an increase in the number of total outpatient visits was associated with a lower hospitalizations (*p* = 0.006), but a higher odds of ER visits (*p* = 0.023). Our findings suggest that policy makers need to adopt systematic policies that focus on restructuring and balancing the structure of resources and service utilization in the three-tier healthcare system.

## 1. Introduction

With a total population of 1.41 billion in 2021, China also has the largest population of elderly and chronically diseased people in the world [1]. At the same time, the economic development and geographical distribution of populations vary widely in China. Thus, establishing an affordable, accessible, and attainable healthcare system is a goal of health development in China. The healthcare system caters to health needs of the elderly mainly through primary care, specialist outpatient, and inpatient services. International evidence has shown that primary care is cheaper, improves health status and efficiency, increases patient satisfaction, and reduces costs [2,3,4]. Consequently, primary care has been promoted internationally as an effective way of providing safe, equitable, and high-quality healthcare. The World Health Organization (WHO) has advocated that every country should establish primary care to be the hub of the healthcare system in order to achieve universal health coverage [5]. 

In 2009, China implemented a new healthcare reform, with a strong commitment to strengthen primary care. Between 2009 and 2020, government subsidies to primary care providers increased from ¥ 19 billion [6] to ¥ 249 billion [7], and the proportion of total government subsidies to total primary care income rose from 12.3–33.1% [6,7]. Furthermore, the government established a universal health insurance system [8], with nearly 97% of population coverage [9], and a national essential drug system, all of which have improved the accessibility and affordability of primary care [9,10]. Our research team previously conducted a population level-based study on the relationship between primary care intensity, health status, and medical expenditures in Guangdong Province and China [11,12,13,14]. These studies revealed the contribution of primary care to the health system in the context of China and provided evidence to support the implementation of policies to strengthen primary care in China.

With the rapid growth in economic levels and aging, healthcare service utilization in China has increased from 2010 to 2021 [15,16]. The annual number of outpatient visits increased from 5.52 to 8.47 billion, with an average annual growth rate of 4.38%. The number of annual admissions increased from 142 to 247 million, with an average annual growth rate of 5.69%. The annual number of ER visits increased from 94.47 to 198.21 million, with an average annual growth rate of 7.69%. The annual hospitalization rate from 15.3–17.5%, which is a lot higher than in the US, the UK, Korea, and Japan [17]. The rapid growth in the utilization of expensive healthcare services, such as hospital inpatient and emergency services, highlights an urgent need to adjust and optimize the structure of health service utilization.

Although China’s new healthcare reform has been paying attention to the development of the primary care system since 2009, it is noteworthy that healthcare services and resources have not been expanded to the primary care system. From 2010 to 2021 [15,16], the proportion of primary care outpatient visits to total outpatient visits decreased from 61.87% to 50.18%. The number of physicians in primary care institutions to the total number of physicians decreased from 32.57% to 29.37%. In fact, the Chinese healthcare system is rapidly moving towards a hospital-oriented trend. Total health expenditures increased from ¥ 1.998 trillion in 2010 [16] to ¥ 7.559 trillion in 2021 [15], with an average annual growth of 15.34%, much higher than average annual gross domestic product (GDP) growth rate of 8.11% over the same period. It has become a key challenge for the healthcare system to control medical cost and improve sustainability in China [18]. 

China has a three-tiered healthcare delivery system, from top to bottom, as follows: large tertiary hospitals, secondary hospitals, and primary care institutions [19]. Patients with health problems can directly choose physicians and medical institutions without a referral. In addition, patients can choose the emergency room (ER) service as their first point of contact. It does admit patients even for minor health problems [20,21]. In China, there are two main types of outpatient services for patients to choose from in a self-referral healthcare system when a health demand arises [19,22]. One is to select hospitals as their usual place for outpatient visits, while the other is to select primary care institutions. Patients who select the former type a priori believe that the level of medical technology and range of services in the hospitals are better than those in primary care institutions.

In this study, we are interested in evaluating whether there are differences in healthcare resource consumption between the two types of outpatient service utilization under the same health status and health needs. In particular, we used the individual-level data from four waves (2011, 2013, 2015, and 2018) of the China Health and Retirement Longitudinal Study (CHARLS) to focus on two research questions. The first was whether individuals who select hospitals outpatient service had different ER and hospital inpatient service utilization than those who select primary care outpatient service. The second was whether ER and hospital inpatient service utilization showed a decreasing or an increasing trend with an increase in the total number of outpatient visits. The answers to these questions can provide the basis for the Chinese government to continuously strengthen health policy of primary care.

## 2. Materials and Methods

### 2.1. Assessment Framework

Establishing an assessment framework is necessary to understand the role of the primary care system in healthcare system performance. We constructed our assessment framework (Figure 1) using the WHO Performance Framework (2000) [23] and the framework for measuring primary care developed by Starfield [24,25,26]. 

Resources were generated from three subsystems—public health, primary care, and hospital care systems. Public health systems strengthen the primary care system by enlarging the coverage of clinical preventive services. Meanwhile, primary care and hospital care systems were linked using a referral system. According to the National Center for Health Statistics in China, healthcare providers in the primary care system include community healthcare centers, township hospitals, healthcare posts, and village clinics/private clinics. In the hospital care system, healthcare providers include general hospitals, specialized hospitals, and Chinese medicine hospitals. The total resources of hospital care and primary care systems, including the number and scale of the facilities, their geographical accessibility, formed the characteristic of the medical care provision, which has a great impact on the performance of healthcare system.

Disease-centered care is provided in a hospital-oriented healthcare system. A strong primary care system provides core values of first contact, accessibility, continuity, comprehensiveness, coordination, and patient-centered care. As health needs emerge, two types of outpatient service utilization often exist when individuals seek medical care. One uses hospital care as the usual source of care (USC) [27,28,29], and the other uses primary care as the USC. Common health problems of individuals can always be satisfied at USCs. When urgent health and major disease problems arise, individuals will also use the ER and hospital inpatient services. This demand will increase when the health needs are not satisfied by the USCs. 

The World Health Report 2008 pointed out that using primary care as the USC had better safety, access, effectiveness, efficiency, user experience, and equity, further leading to better health gains, equity in health, financial protection, equity in finance, responsiveness, and equity in responsiveness to the health system [30]. The availability and characteristics of health resources affect the individuals’ selection of the USC. Internationally, the common direction of health reforms in all countries is to strengthen the primary care system, make it the hub of the healthcare system, and reverse hospital-oriented trends in healthcare [30]. 

Our study focused on healthcare delivery and utilization systems, which are in the dashed boxes of Figure 1. We will explore the association between different types of outpatient service utilization and ER or hospital inpatient service utilization.

### 2.2. Data Sources

We used individual-level panel data from the four waves of the latest available data (2011, 2013, 2015, and 2018) of the China Health and Retirement Longitudinal Study (CHARLS). The national baseline survey of CHARLS was conducted between June 2011 and March 2012, with subsequent follow-up every 2 years, and a total of four waves are currently being updated. The CHARLS study adopted a stratified, multi-stage probability-proportionate-to-size (PPS) sampling strategy. In the first stage, all counties/districts in China (except for Tibet) were stratified by region, within region by rural counties or urban districts, and by per capita GDP. Subsequently, 150 counties/districts were randomly selected by PPS sampling. In the next stage, in each county/district, three villages/communities were randomly chosen using PPS sampling. Thus, this study was conducted in 28 provinces, 150 countries/districts, and 450 villages/urban communities across China. In the third stage, a dedicated mapping software (CHARLS-GIS) designed and developed by CHARLS was used to conduct field mapping and produce a list of dwelling units in each village/community, from which a number of dwellings were then randomly selected. If there was more than one household meeting the age eligibility in a given dwelling, one household was randomly selected. Finally, in each sampled household, a short screening form from the household questionnaire was used to screen out whether a member met the age eligibility requirement in the household. If there were members aged >45 years in a household, one member was randomly selected; a selected member aged > 45 years was automatically chosen as the main respondent [31]. The core household questionnaire includes the following sections: demographics; family structure/transfer; health; health insurance and health care utilization; work, retirement and pension; income, expenditures and assets; housing characteristics and the community and policy modules. To ensure the standardization of the interviews and the accuracy of the data, CHARLS uniformly provided rigorous training to the recruited university students and sent them to conduct field surveys across the country. During the fieldwork, each respondent who agreed to participate in the survey had to sign two informed consent forms. The Institutional Review Board at Peking University granted ethical approval for this study (approval number is IRB00001052-11015). We performed secondary data analysis in this study. 

The numbers of respondents in the survey were 17,705, 18,605, 21,095, and 19,816, for the 2011, 2013, 2015, and 2018 waves, respectively. A sample of 44,859 participants were obtained by excluding the missing values (12,645, 13,490, 7031, and 11,687 respondents in 2011, 2013, 2015, and 2018, respectively). Based on our research question, respondents using outpatient services were selected. We defined two types of outpatient visit, as follows: one is primary care outpatient visits, included respondents who selected community healthcare centers, township hospitals, healthcare posts, and village clinics/private clinics, and the other is hospital outpatient visits, included respondents who selected general hospitals, specialized hospitals, and Chinese medicine hospitals as outpatient healthcare providers in the previous month. Among the respondents, 276 who selected both primary and hospital outpatient care in the previous month were excluded, as this number was relatively small for reliable analysis [32]. Finally, a total of 7544 respondents (16.82% of the total population sample) were included in this study (2190, 2463, 1214, and 1677 respondents, respectively, in the 2011, 2013, 2015, and 2018 waves). The final dataset includes four waves of respondents to form a pooled cross-section data. 

### 2.3. Variables

#### 2.3.1. Dependent Variables

Dependent variables were the ER and hospital inpatient service utilization. Whether respondents had ER visits during the previous month served as a proxy for ER service utilization [28,33,34]. Whether the respondent received hospital inpatient care, the number of hospitalizations, and the length of hospitalization days in the previous year were used as proxies for hospital inpatient service utilization.

#### 2.3.2. Exposure

Outpatient service utilization during the previous month was the exposure in this study. The outpatient visit types, including primary care and hospital outpatient visits, was used as a proxy of the outpatient service utilization.

#### 2.3.3. Covariates

Factors including the availability and characteristics of health resources, health status, demographic characteristics, health needs, enabling factors, and family support influenced the selection and frequency of health services [34,35,36,37,38,39]. We, therefore, included current residence, GDP per capita (PGDP at prefecture-city-level), and economic region [40] (province-level), self-reported health status, activities of daily life (ADLs) limitations, instrumental activities of daily life (IADLs) limitations, depression, age, sex, marital status, education, work status, drinking, smoking, number of chronic diseases, frequency of social activity, medical insurance status, per capita household consumption expenditure (PCE), number of caregivers, and living arrangements as covariates. The definitions and assignments of all variables are shown in Table A1.

### 2.4. Statistical Analysis

Descriptive statistics for ER, hospital inpatient, and outpatient service utilization (and covariates) were reported as the mean ± standard deviation (SD) for continuous variables, and the frequency (N) and percentage (%) for categorical variables in each year and overall. The chi-square tests, two-sample Student’s t-test, and one-way analysis of variance (ANOVA) were used to test the differences in the covariates between the outpatient visit types. The chi-square and Mann–Whitney *U* tests were used to explore the differences in the ER and hospital inpatient services utilization between the outpatient visit types.

We used a two-part model to analyze ER and hospital inpatient services utilization. In the first part, we examine the impact of outpatient visit types on ER visits and hospitalization (yes or no). In the second part, we used zero-truncated negative binomial regression to explore the relationship of the length of hospitalization and the number of hospitalizations with outpatient visit types, conditional on at least one hospitalization. The year was treated as a fixed effect in all models to control for unmeasured time-variant characteristics of services utilization. The model equation for negative binomial regression can be written as follows:
$$\mathrm{In}\;Y=\beta 0\;+\;\beta_{1}X_{1}\;+\sum_{j=2}^{k}\beta_{j}X_{j}\;+\;\delta T_{t}$$
where Y represents the number of hospitalizations or the length of hospitalization in days, *X*_1_ represents the independent variable that is the outpatient visit types, *X_j_* represents all covariates, *j* is the number of covariates, and T_t_ is the vector of the year dummy variables. Here, β_0_ is the constant term, β_1_ is the negative binomial regression coefficient of the exposure, β*_j_* (*j* = 2, …, k) is the vector of regression coefficients of *X_j_*, and δ is the coefficient of the year dummy variables comparing with the reference year. Negative binomial regression was used because the variance of dependent variables was highly inflated. Zero-truncated negative binomial regression was used by excluding the non-hospitalized respondents from the analyses.

Stratified analyses were performed by classifying the total number of outpatient visits in the previous month into three subgroups (1, 2, and ≥3 times), using the above-mentioned regression analysis to explore the association between the outpatient visit types and ER or hospital inpatient service utilization in each subgroup. Furthermore, a trend test was applied to explore whether there was a trend in the association between the outpatient visit types and the odds of ER visit or the number of hospitalizations as the total number of outpatient visits increased.

Adjusted odds ratios (ORs), adjusted incidence rate ratios (IRRs), average marginal effect with their *p*-values, and robust standard errors were reported. Statistical analyses were performed using Stata MP version 16, and a *p* < 0.05 was considered statistically significant. Forest plots were mapped using GraphPad Prism 9.0. (GraphPad Software, San Diego, CA, USA).

## 3. Results

### 3.1. Descriptive Statistics 

Table 1 and Table 2 shows the descriptive analysis of the covariates in 2011, 2013, 2015, 2018. A total of 7754 respondents were included. Most of the survey respondents were aged 50–69 years (68.85%) and were evenly distributed across the country (32.37% in Western China, 36.35% in Middle China, and 31.28% in Eastern China). Most of the respondents were women (59.04%), with a spouse (84.19%), and lived in rural areas (64.59%). Nearly half of them were employed (49.85%), and the majority had an educational level of no higher than elementary school (70.16%). From 2011 to 2018, both PCE and PGDP displayed a steady upward trend, medical insurance coverage increased from 93.29% in 2011 to 96.54% in 2018, and those reporting a good health status decreased from 31.92% to 10.56%, as confirmed by a steady increase in the proportion of respondents with multiple chronic diseases. More importantly, there was no statistically significant difference in self-reported health status between primary care and hospital outpatient visit respondents, which implies that respondents’ use of different types of outpatient services (primary care vs. hospital) was not associated with respondents’ health status.

As shown in Figure 2, from 2011 to 2018, the rate of primary care outpatient visits decreased, whereas those using hospital outpatient services increased. The rate of ER visits and hospitalization increased biennially during this period, from 2.37% to 4.05%, and from 14.11% to 27.01%, respectively. The mean number of hospitalizations and length of hospitalization in the previous year increased from 0.21 to 0.48 times and 1.63 to 3.28 days, respectively (data not shown). 

Table 3 presents the descriptive statistics of ER and hospital inpatient services utilization by two outpatient visit types in the three subgroups and overall. In the total sample, the rate of ER visits and hospitalization, the mean number of hospitalizations, and length of hospitalization in days were significantly lower for respondents with primary care outpatient visits than in those with hospital outpatient visits. Consistent significant differences were noted in the three subgroups.

### 3.2. Association between Outpatient Visits Types and ER or Hospital Inpatient Services Utilization

Table 4 shows the results of the association between the outpatient visit types and the utilization of ER or hospital inpatient services, after controlling for all covariates. Overall, respondents with primary care outpatient visits were less likely to have ER visits (adjusted OR = 0.141, 95% confidence interval [CI]: 0.101–0.194) and hospitalization (adjusted OR = 0.623, 95% CI: 0.546–0.711) than those reporting hospital outpatient visits. Among respondents with at least one hospitalization, those with primary care outpatient visits had fewer hospitalization days (adjusted IRR = 0.886, 95% CI: 0.810–0.969) than those reporting hospital outpatient visits.

### 3.3. Trend Test of Association between Outpatient Visits Types and ER or Hospital Inpatient Services Utilization

Table 5 shows the results of the analysis with respondents stratified based on the total number of outpatient visits. We consistently found that, in the three sub-groups, respondents with primary care outpatient visits were less likely to have ER visits (*p* < 0.001) and hospitalization (*p* < 0.05) than those with hospital outpatient visits. Additionally, among respondents with at least one hospitalization, those with one primary care outpatient visit a month had fewer days in hospital (adjusted IRR = 0.828, 95% CI: 0.740–0.927) than those reporting one hospital outpatient visit a month. Furthermore, an increasing number of total outpatient visits were associated with higher odds of ER visits (*p* for trend = 0.023), and a lower number of hospitalizations (*p* for trend = 0.006). 

Figure 3 is a forest plot illustrating the above associations. Logistic regressions show significant negative associations between the outpatient visit types and ER visits (adjusted OR = 0.131–0.173) or hospitalization (adjusted OR = 0.577–0.724), either in the overall group or in the three subgroups. Using the zero-truncated negative binomial regression, there is no statistically significant association between the outpatient visit types and the number of hospitalizations (*p* > 0.500). However, there was a significant negative association between the outpatient visit types and the length of hospitalization in days for inpatients (*p* = 0.008).

## 4. Discussion

### 4.1. Main Findings

This study used four waves (2011, 2013, 2015, and 2018) of data from CHARLS, a national representative community-based household survey, and explored the association between outpatient visit types and ER or hospital inpatient services utilization among middle-aged and elderly individuals (aged 45 and above) in self-referral system in China. This study identifies a decreasing temporal trend in primary care outpatient visits rates, and an increasing trend in hospital outpatient visits rates in China from 2011 to 2018. It suggests that, during this period of 2011 to 2018, patients in China tended to depend more and more on large hospitals through the self-referral system, which showed that China’s healthcare system is rapidly moving towards a hospital-oriented mode. This standing trend further warranted our exploration of the usage of primary care in China’s healthcare system. The following are the two main findings of this study: first, we found that compared to respondents who had hospital outpatient visits, those who had primary care outpatient visits had significantly lower odds of ER visits and hospitalization, and fewer hospitalization days. Second, the trend test indicated that an increasing number of total outpatient visits was significantly associated with a lower number of hospitalizations and a slightly greater chance of ER visits. To the best of our knowledge, the findings of the trend test add new evidence to the value of primary care.

### 4.2. Association between Outpatient Visits Types and ER or Hospital Inpatient Services Utilization

This study showed that compared with respondents who select a hospitals’ outpatient service, those who select a primary care outpatient service are significantly associated with lower ER and hospital inpatient services utilization. Moreover, we also conducted a stratified analysis by respondents’ self-reported health status and found consistent results (Table A2). It showed that the association between the primary care usage and ER or inpatient usage was not modified by the health status. Moreover, in self-referral system in China, ERs do not only treat critically illed patients, but patients with minor health problems may also use ERs. This somehow weakens the association between ER usage and the severity of the patient’s disease.

The first main finding is consistent with existing findings of international studies. Bertakis et al. recruited 509 patients who had outpatient appointments at a Medical Center and randomly assigned them into family practice and internal medicine clinics and found that family practice patients were significantly less likely to visit the emergency department [34], and also had a shorter average length of stay [41] than internal medicine clinics’ patients. Greenfield sampled physicians from different specialties in three US states and analyzed differences in healthcare service utilization among their visiting patients, and observed that hospitalization rates were 100% and 50% higher for patients visiting cardiologists and endocrinologists, respectively, than patients visiting family doctors [29]. Sung et al. used Korea’s national representative household data to evaluate the association between the types of USCs and emergency or hospitalization visits, and found that respondents who had primary care physicians as USCs had lower odds of emergency visits and hospitalization, while respondents who had hospital specialists as USCs had higher odds of hospitalization [28]. Fung et al. used population-based data to explore the differences in healthcare service utilization and patterns among patients visiting different types of primary care physicians, and found that residents in Hong Kong, China with a regular family doctor had less likelihood of emergency and inpatient visits than of those in the “other regular doctor” or “no regular doctor” groups [20]. Huang et al. used Taiwan’s National Health Insurance Database to conduct a population-based retrospective cohort study, and observed that a Family Physician Integrated Care Program policy could reduce hospital admission in the long term in Taiwan [21].

The practice styles between primary care physicians and hospital specialists may explain why primary care might reduce ER and hospital inpatient services utilization. Hospital specialist outpatient care is more disease-centered care, emphasizing disease treatment rather than prevention, curing diseases rather than curing human beings, and, as such, it is more likely to recommend expensive drugs, laboratory tests, and inpatient care [18]. However, the practice style of primary care lies in its core values, namely first-contact care, accessibility, continuity, comprehensiveness, coordination, and patient-centered care. First-contact care provides a formal starting point for patients to enter the healthcare system [42]. General practitioners act as “gatekeepers” to solve common health problems, particularly chronic diseases, and are less likely to need appointments [43,44]. Therefore, patients receive timely, effective outcomes for their health problems and are less likely to experience barriers to accessing care than with hospital specialist care [45,46,47]. As primary care focuses on prevention, general practitioners may prevent the continued escalation of medical problems through early detection and treatment [28,48]. During consultation, physicians can reduce patients’ anxiety and improve their trust [49] by increasing patient engagement [50]. Thus, patients are willing to actively provide accurate information to physicians. The accumulation of patient information means improved understanding of patient preferences [51], reduced counseling time [52], reduced diagnostic uncertainty [50], and increased patient satisfaction and adherence to treatment regimens, thus, facilitating the management of chronic diseases and reducing inpatient care [53,54,55,56,57]. Furthermore, with more complicated and serious diseases, general practitioners use their professional knowledge to promptly screen and refer the patient. Coordinated care by general practitioners provides patients with more cost-effective care, resulting in fewer hospital services being utilized [58,59,60].

### 4.3. Trend Test of Association between Outpatient Visits Types and ER or Hospital Inpatient Services Utilization

There are two main trend findings, as follows: firstly, a decreasing trend in the number of hospitalizations, and secondly, an increasing trend in the chance of ER visits with the number of total outpatient visits. The finding of the first trend can be explained by the following assumption: as the number of outpatient visits increased, the general practitioners had increased contact with the patients, resulting in a stronger relationship. This long-term, stable relationship enabled general practitioners to continuously accumulate patient information, created a more comprehensive understanding of patients, facilitated the identification of various patient needs, as well as provided comprehensive services for patients, and provided referrals to more appropriate secondary or tertiary care facilities [20,61,62]. A better relationship between general practitioners and patients would stimulate the interaction mechanism between continuity of services and other core values of primary care, and these core values may reinforce each other, resulting in cumulative benefits for patients, which may reduce hospital inpatient services utilization [2]. However, it is harder to keep the long-term relationship with providers in hospitals than with those in primary care clinics. In response to the explanation of the finding of the second trend, we argue that the more frequently a patient chooses outpatient services at a certain provider or clinic, the more likely the patient select that provider or clinic as their USC [63]. Patients who select hospital care as their USC were more likely to seek ER services directly at the hospital once they had a health problem. Therefore, in terms of reducing the ER visits, patients would benefit from primary care outpatient services rather than hospital care outpatient services. However, more frequent primary care outpatient services may not be conducive to a greater reduction in ER services utilization; they would benefit most at the frequency of one visit per month.

### 4.4. Study Limitations

This study has some limitations. Firstly, this was a correlational study and, thus, we could not examine causation. Secondly, our study only included middle-aged and elderly respondents, thus, limiting the generalization of the conclusion to people not within the age range of the study.

## 5. Conclusions

In the context of a hospital-oriented self-referral healthcare system, for middle-aged and elderly individuals (aged 45 years and above), those who select primary care outpatient service had significantly lower ER and hospital inpatient services utilization relative to those who select a hospitals’ outpatient service. More importantly, as the total number of outpatient visits increased, primary care respondents tended to show a lower number of hospitalizations and higher odds of ER visits compared to those who had the same number of hospital outpatient visits. Our findings provide robust evidence for the policy that government continues to strengthen the primary care system. Moreover, it further suggests that, in China, strengthening the primary care system by itself is not enough to achieve the transformation of the healthcare system to a system centered on primary care. The administrators and stakeholders are suggested to adopt systematic policies and measures that focus on restructuring and balancing the structure of resources and service utilization in the three-tier healthcare system.

## Figures and Tables

**Figure 1 ijerph-19-12979-f001:**
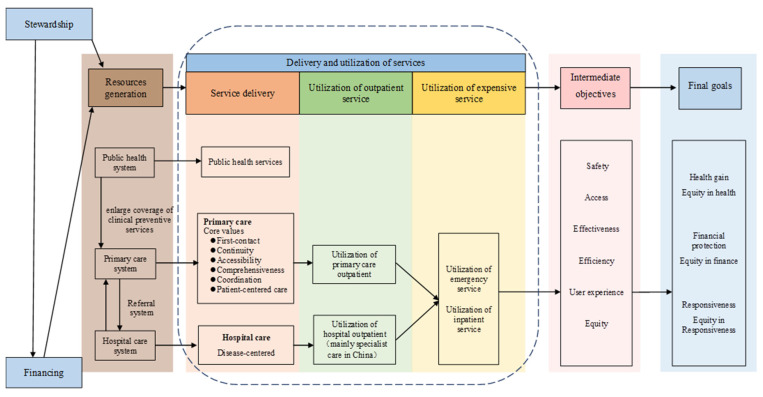
The assessment framework.

**Figure 2 ijerph-19-12979-f002:**
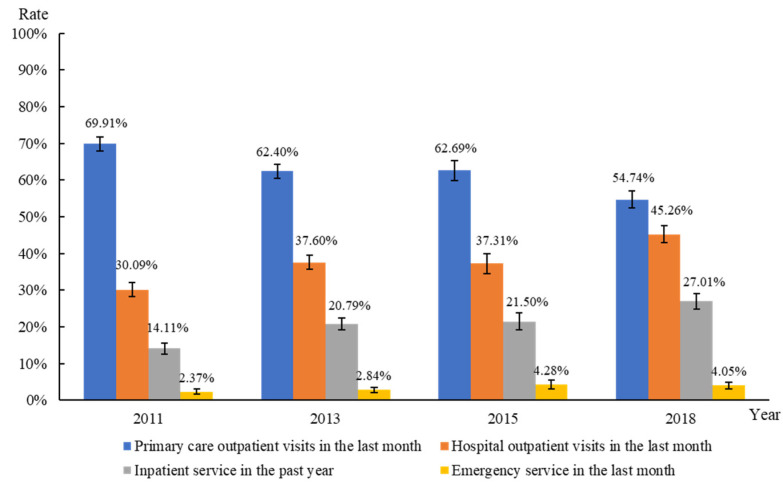
Outpatient services, ER services, and hospital inpatient services utilization from 2011 to 2018.

**Figure 3 ijerph-19-12979-f003:**
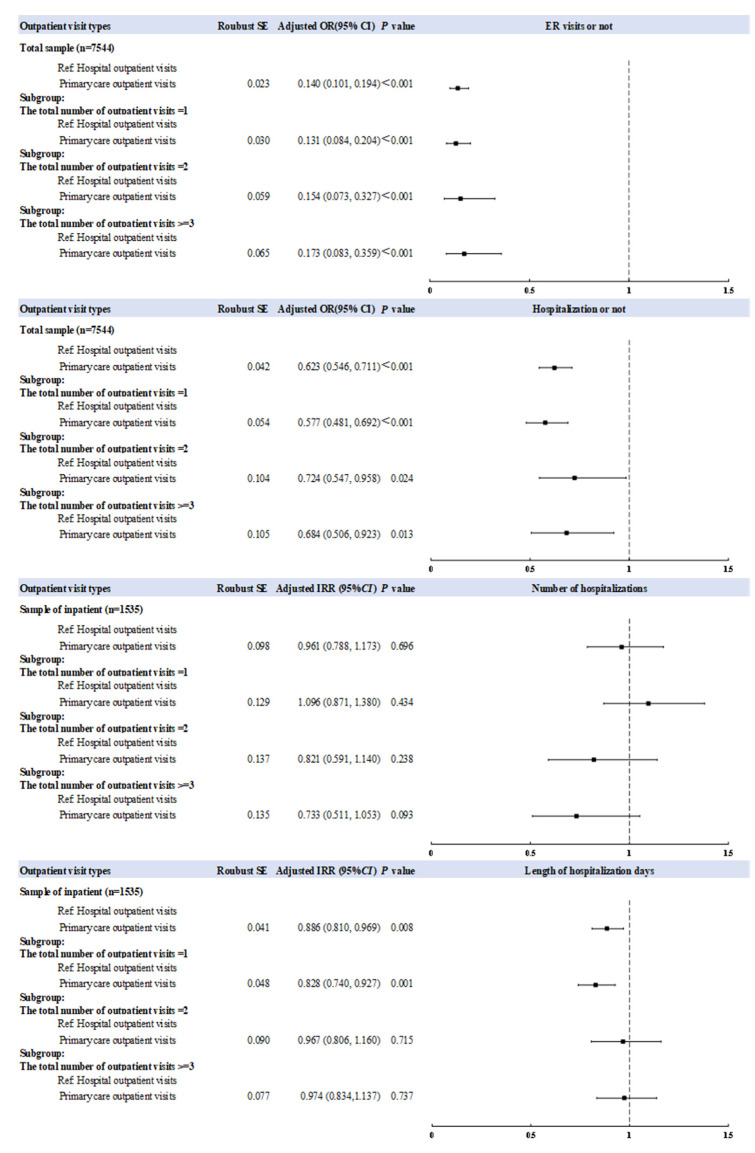
Forest plot of the association between the outpatient visit types and ER or hospital inpatient utilization. All models were adjusted for economic region, PGDP, current residence, self-reported health status, ADL limitations, IADL limitations, depression, age group, sex, marital status, education, work status, smoking, drinking, number of chronic diseases, frequency of social activity, medical insurance status, PCE, number of caregivers, and living arrangement.

**Table 1 ijerph-19-12979-t001:** Descriptive statistics of covariates in each wave, 2011–2018.

		2011 (*n* = 2190)		2013 (*n* = 2463)		2015 (*n* = 1214)		2018 (*n* = 1677)	
		Hospital Outpatient	Primary Care Outpatient	Hospital Outpatient	Primary Care Outpatient	Hospital Outpatient	Primary Care Outpatient	Hospital Outpatient	Primary Care Outpatient
Economic region (%)	West	187 (28.38)	498 (32.53)	252 (27.21)	556 (36.17) ^$^	141 (31.13)	260 (34.17)	229 (30.17)	319 (34.75)
	Middle	259 (39.30)	563 (36.77)	319 (34.45)	577 (37.54)	160 (35.32)	263 (34.56)	292 (38.47)	309 (33.66)
	East	213 (32.32)	470 (30.70)	355 (38.34)	404 (26.28)	152 (33.55)	238 (31.27)	238 (31.36)	290 (31.59)
PGDP, mean (SD)		36,982.30 (21,940.17)	32,471.47 (19,918.12) ^$ #^	49,101.86 (29,244.55)	39,804.78 (24,459.99) ^$^	51,337.25 (31,650.07)	45,033.61 (27,548.94) ^$^	59,428.61 (34,705.76)	51,235.13 (29,271.44) ^$^
Current residence, *n* (%)	Rural	341 (51.75)	1099 (71.78) ^$ #^	480 (51.84)	1056 (68.71) ^$^	254 (56.07)	554(72.80) ^$^	415 (54.68)	674 (73.42) ^$^
	Urban	318 (48.25)	432 (28.22)	446 (48.16)	481 (31.29)	199 (43.93)	207 (27.20)	344 (45.32)	244 (26.58)
Self-reported health status, *n* (%)	Poor	313 (47.50)	717 (46.83) ^#^	375 (40.50)	602 (39.17)	196 (43.27)	313 (41.13)	329 (43.4)	410 (44.66)
	Fair	143 (21.70)	318 (20.77)	225 (24.30)	382 (24.85)	100 (22.08)	179 (23.52)	346 (45.65)	414 (45.10)
	Good	203 (30.80)	496 (32.40)	326 (35.21)	553 (35.98)	157 (34.66)	269 (35.35)	83 (10.95)	94 (10.24)
ADL limitations, *n* (%)	No	516 (78.30)	1130 (73.81) ^$ #^	714 (77.11)	1152 (74.95)	341 (75.28)	541(71.09)	576 (75.89)	612 (66.67) ^$^
	Yes	143 (21.70)	401 (26.19)	212 (22.89)	385 (25.05)	112 (24.72)	220 (28.91)	183 (24.11)	306 (33.33)
IADL limitations, *n* (%)	No	477 (72.38)	1067 (69.69) ^#^	682 (73.65)	1109 (72.15)	319 (70.42)	547 (71.88)	536 (70.62)	574 (62.53) ^$^
	Yes	182 (27.62)	464 (30.31)	244 (26.35)	428 (27.85)	134 (29.58)	214 (28.12)	223 (29.38)	344 (37.47)
Depression, *n* (%)	No	347 (52.66)	681 (44.48) ^$ #^	560 (60.48)	871 (56.67)	269 (59.38)	403 (52.96) ^$^	422 (55.60)	416 (45.32) ^$^
	Yes	312 (47.34)	850 (55.52)	366 (39.52)	666 (43.33)	184 (40.62)	358 (47.04)	337 (44.40)	502 (54.68)
Age group, *n* (%)	45–49	97 (14.72)	197 (12.87) ^$ #^	104 (11.23)	182 (11.84)	21 (4.64)	40 (5.26)	2 (0.26)	2 (0.22)
	50–59	249 (37.78)	494 (32.27)	345 (37.26)	505 (32.86)	157 (34.66)	247 (32.46)	281 (37.02)	288 (31.37)
	60–69	207 (31.41)	527 (34.42)	293 (31.64)	541 (35.20)	169 (37.31)	286 (37.58)	250 (32.94)	355 (38.67)
	70–79	86 (13.05)	247 (16.13)	145 (15.66)	244 (15.88)	89 (19.65)	139 (18.27)	182 (23.98)	209 (22.77)
	≥80	20 (3.03)	66 (4.31)	39 (4.21)	65 (4.23)	17 (3.75)	49 (6.44)	44 (5.8)	64 (6.97)
Sex, *n* (%)	Female	372 (56.45)	923 (60.29)	563 (60.80)	914 (59.47)	228 (50.33)	454 (59.66) ^$^	429 (56.52)	571 (62.20) ^$^
	Male	287 (43.55)	608 (39.71)	363 (39.20)	623 (40.53)	225 (49.67)	307 (40.34)	330 (43.48)	347 (37.80)
Marital status, *n* (%)	Without spouse	75 (11.38)	219 (14.30) ^#^	130 (14.04)	226 (14.70)	95 (20.97)	182 (23.92)	101 (13.31)	165 (17.97) ^$^
	With spouse	584 (88.62)	1312 (85.70)	796 (85.96)	1311 (85.30)	358 (79.03)	579 (76.08)	658 (86.69)	753 (82.03)
Education, *n* (%)	Elementary school and below	406 (61.61)	1208 (78.90) ^$ #^	536 (57.88)	1155 (75.15) ^$^	254 (56.07)	572 (75.16) ^$^	446 (58.76)	716 (78.00) ^$^
	Secondary school	230 (34.90)	313 (20.44)	347 (37.47)	370 (24.07)	175 (38.63)	186 (24.44)	290 (38.21)	198 (21.57)
	College and above	23 (3.49)	10 (0.65)	43(4.64)	12 (0.78)	24 (5.30)	3 (0.39)	23 (3.03)	4 (0.44)
Work status, *n* (%)	Unemployed	303 (45.98)	594 (38.80) ^$ #^	382 (41.25)	508 (33.05) ^$^	175 (38.63)	273 (35.87) ^$^	279 (36.76)	306 (33.33) ^$^
	Employed	227 (34.45)	870 (56.83)	335 (36.18)	913 (59.40)	160 (35.32)	432 (56.77)	281 (37.02)	543 (59.15)
	Retired	129 (19.58)	67 (4.38)	209 (22.57)	116 (7.55)	118 (26.05)	56 (7.36)	199 (26.22)	69 (7.52)
Smoking, *n* (%)	Never	435 (66.01)	998 (65.19) ^$ #^	618 (66.74)	987 (64.22) ^$^	250 (55.19)	480 (63.07) ^$^	469 (61.79)	570 (62.09)
	Used to smoke	94 (14.26)	138 (9.01)	111 (11.99)	132 (8.59)	99 (21.85)	110 (14.45)	146 (19.24)	146(15.90)
	Now	130 (19.73)	395 (25.80)	197 (21.27)	418 (27.20)	104 (22.96)	171 (22.47)	144 (18.97)	202 (22.00)
Drinking, *n* (%)	Never	488 (74.05)	1097 (71.65)	694 (74.95)	1090 (70.92) ^$^	298 (65.78)	537 (70.57)	551 (72.60)	670 (72.98) ^$^
	<1 time per month	40 (6.07)	118 (7.71)	74 (7.99)	120 (7.81)	42 (9.27)	56 (7.36)	63 (8.30)	49 (5.34)
	≥1 times per month	131 (19.88)	316 (20.64)	158 (17.06)	327 (21.28)	113 (24.94)	168 (22.08)	145 (19.1)	199 (21.68)
Number of chronic diseases, *n* (%)	0	101 (15.33)	283 (18.48) ^#^	133 (14.36)	265 (17.24)	61 (13.47)	128 (16.82) ^$^	43 (5.67)	77 (8.39)
	1	179 (27.16)	420 (27.43)	238 (25.70)	419 (27.26)	84 (18.54)	174 (22.86)	123 (16.21)	147 (16.01)
	≥2	379 (57.51)	828 (54.08)	555 (59.94)	853 (55.50)	308 (67.99)	459 (60.32)	593 (78.13)	694 (75.6)
Frequency of social activity, *n* (%)	None	321 (48.71)	762 (49.77) ^#^	366 (39.52)	625 (40.66)	172 (37.97)	317 (41.66)	305 (40.18)	422 (45.97) ^$^
	Not regular	89 (13.51)	224 (14.63)	129 (13.93)	238 (15.48)	74 (16.34)	130 (17.08)	117 (15.42)	144 (15.69)
	Almost every week	70 (10.62)	180 (11.76)	108 (11.66)	191 (12.43)	61 (13.47)	93 (12.22)	88 (11.59)	112 (12.20)
	Almost every day	179 (27.16)	365 (23.84)	323 (34.88)	483 (31.42)	146 (32.23)	221 (29.04)	249(32.81)	240 (26.14)
Medical insurance status, *n* (%)	None	23 (3.49)	79 (5.16) ^$ #^	34 (3.67)	36 (2.34) ^$^	19 (4.19)	27 (3.55) ^$^	11 (1.45)	18 (1.96) ^$^
	Urban Employee Basic Medical Insurance	112 (17.00)	56 (3.66)	198 (21.38)	91 (5.92)	98 (21.63)	57 (7.49)	154 (20.29)	49 (5.34)
	Urban and rural resident medical insurance	469 (75.27)	1379 (90.07)	664 (71.17)	1400 (91.09)	313 (69.09)	667 (87.65)	570 (75.10)	846 (92.16)
	other medical insurance	28 (4.25)	17 (1.11)	30 (3.24)	10 (0.65)	23 (5.08)	10 (1.31)	24 (3.16)	5 (0.54)
PCE/yuan, mean (SD)		4070.36 (3010.56)	2793.09 (2161.98)	3138.75 (2289.70)	2222.34 (1665.37)	6372.15 (5190.50)	4487.86 (3703.05)	8878.39 (6564.50)	5851.12 (4792.40)
ln PCE, mean (SD)		8.04 (0.77)	7.67 (0.73) ^$ #^	7.80 (0.72)	7.46 (0.72) ^$^	8.44 (0.82)	8.13 (0.74) ^$^	8.82 (0.77)	8.39 (0.76) ^$^
Number of caregivers, *n* (%)	0	525 (79.67)	1245 (81.32) ^#^	742 (80.13)	1212 (78.85)	344 (75.94)	580 (76.22)	575 (75.76)	637 (69.39) ^$^
	1	13 (1.97)	38 (2.48)	105 (11.34)	193 (12.56)	54 (11.92)	67 (8.80)	87 (11.46)	129 (14.05)
	2–3	108 (16.39)	208 (13.59)	61 (6.59)	103 (6.70)	30 (6.62)	69 (9.07)	51 (6.72)	72 (7.84)
	≥4	13 (1.97)	40 (2.61)	18 (1.94)	29 (1.89)	25 (5.52)	45 (5.91)	46 (6.06)	80 (8.71)
Living arrangement, *n* (%)	Alone	28 (4.25)	108 (7.05) ^$ #^	95 (10.26)	153 (9.95)	66 (14.57)	110 (14.45)	86 (11.33)	117 (12.75)
	With relatives	32 (4.86)	118 (7.71)	48 (5.18)	120 (7.81)	31 (6.84)	78 (10.25)	48 (6.32)	34 (3.70)
	With spouse	568 (86.19)	1218 (79.56)	768 (82.94)	1237 (80.48)	186 (41.06)	286 (37.58)	413 (54.41)	486 (52.94)
	With offspring	31 (4.70)	87 (5.68)	15 (1.62)	27 (1.76)	170 (37.53)	287 (37.71)	212 (27.93)	281 (30.61)

Here, PCE = the per capita household consumption expenditure; ln PCE = natural logarithm of the per capita household consumption expenditure; PGDP = gross domestic product (GDP) per capital at prefecture city level. ^$^ There are differences in covariates across the outpatient visit types based on the Chi-square test, t-test, and one-way ANOVA. ^#^ There are differences in covariates across the years based on the Chi-square test, t-test, and one-way ANOVA.

**Table 2 ijerph-19-12979-t002:** Descriptive statistics of covariates in all four waves (N = 7544).

		Hospital Outpatient	Primary Care Outpatient	*p* Value	Total
Economic region (%)	West	809 (28.92)	1633 (34.40)	<0.001	2442 (32.37)
	Middle	1030 (36.83)	1712 (36.06)		2742 (36.35)
	East	958 (34.25)	1402 (29.53)		2360 (31.28)
PGDP, mean (SD)		49,410.71 (30,805.59)	40,488.35 (25,595.87)	<0.001	43,796.39 (27,974.17)
Current residence, *n* (%)	Rural	1490 (53.27)	3383 (71.27)	<0.001	4873 (64.59)
	Urban	1307 (46.73)	1364 (28.73)		2671 (35.41)
Self-reported health status, *n* (%)	Poor	1213 (43.38)	2042 (43.02)	0.070	3256 (43.15)
	Fair	814 (29.11)	1293 (27.24)		2107 (27.93)
	Good	769 (27.50)	1412 (29.75)		2181 (28.91)
ADL limitations, *n* (%)	No	2147 (76.76)	3435 (72.36)	<0.001	5582 (73.99)
	Yes	650 (23.24)	1312 (27.64)		1962 (26.01)
IADL limitations, *n* (%)	No	2014 (72.01)	3297 (69.45)	0.019	5311 (70.40)
	Yes	783 (27.99)	1450 (30.55)		2233 (29.60)
Depression, *n* (%)	No	1598 (57.13)	2371 (49.95)	<0.001	3969 (52.61)
	Yes	1199 (42.87)	2376 (50.05)		3575 (47.39)
Age group, *n* (%)	45–49	224 (8.01)	421 (8.87)	<0.001	645 (8.55)
	50–59	1032 (36.90)	1534 (32.32)		2566 (34.01)
	60–69	919 (32.86)	1709 (36.00)		2628 (34.84)
	70–79	502 (17.95)	839 (17.67)		1341 (17.78)
	≥80	120 (4.29)	244 (5.14)		364 (4.83)
Age, mean (SD)		61.85(9.55)	62.35(9.56)	0.029	62.17(9.56)
Sex, *n* (%)	Female	1592 (56.92)	2862 (60.29)	0.004	4454 (59.04)
	Male	1205 (43.08)	1885 (39.71)		3090 (40.96)
Marital status, *n* (%)	Without spouse	401 (14.34)	792 (16.68)	0.007	1193 (15.81)
	With spouse	2396 (85.66)	3955 (83.32)		6351 (84.19)
Education, *n* (%)	Elementary school and below	1642 (58.71)	3651 (76.91)	<0.001	5293 (70.16)
	Secondary school	1042 (37.25)	1067 (22.48)		2109 (27.96)
	College and above	113 (4.04)	29 (0.61)		142 (1.88)
Work status, *n* (%)	Unemployed	1139 (40.72)	1681 (35.41)	<0.001	2820 (37.38)
	Employed	1003 (35.86)	2758 (58.10)		3761 (49.85)
	Retired	655 (23.42)	308 (6.49)		963 (12.77)
Smoking, *n* (%)	Never	1772 (63.35)	3035 (63.94)	<0.001	4807 (63.72)
	Used to smoke	450 (16.09)	526 (11.08)		976 (12.94)
	Now	575 (20.56)	1186 (24.98)		1761 (23.34)
Drinking, *n* (%)	Never	2031 (72.61)	3394 (71.50)	0.158	5425 (71.91)
	<1 time per month	219 (7.83)	343 (7.23)		562 (7.45)
	≥1 times per month	547 (19.56)	1010 (21.28)		1557 (20.64)
Number of chronic diseases, *n* (%)	0	338 (12.08)	753 (15.86)	<0.001	1091 (14.46)
	1	624 (22.31)	1160 (24.44)		1784 (23.65)
	≥2	1835 (65.61)	2834 (59.70)		4669 (61.89)
Frequency of social activity, *n* (%)	None	1164 (41.62)	2126 (44.79)	0.001	3290 (43.61)
	Not regular	409 (14.62)	736 (15.50)		1145 (15.18)
	Almost every week	327 (11.69)	576 (12.13)		903 (11.97)
	Almost every day	897 (32.07)	1309 (27.58)		2206 (29.24)
Medical insurance status, *n* (%)	None	87 (3.11)	160 (3.37)	<0.001	247 (3.27)
	Urban Employee Basic Medical Insurance	562 (20.09)	253 (5.33)		815 (10.80)
	Urban and rural resident medical insurance	2043 (73.04)	4292 (90.41)		6335 (83.97)
	other medical insurance	105 (3.75)	42 (0.88)		147 (1.95)
PCE/yuan, mean (SD)		5439.45 (5044.94)	3471.36 (3309.48)		4201.05 (4150.81)
ln PCE, mean (SD)		8.24 (0.87)	7.81 (0.82)	<0.001	7.97 (0.86)
Number of caregivers, *n* (%)	0	2186 (78.16)	3674 (77.40)	0.620	5860 (77.68)
	1	259 (9.26)	427 (9.00)		686 (9.09)
	2–3	250 (8.94)	452 (9.52)		702 (9.31)
	≥4	102 (3.65)	194 (4.09)		296 (3.92)
Living arrangement, *n* (%)	Alone	275 (9.83)	488 (10.28)	0.027	763 (10.11)
	With relatives	159 (5.68)	350 (7.37)		509 (6.75)
	With spouse	1935 (69.18)	3227 (67.98)		5162 (68.43)
	With offspring	428 (15.30)	682 (14.37)		1110 (14.71)

Here, PCE = the per capita household consumption expenditure; ln PCE = natural logarithm of the per capita household consumption expenditure; PGDP = gross domestic product (GDP) per capital at prefecture city level.

**Table 3 ijerph-19-12979-t003:** ER and hospital inpatient services utilization by the outpatient visit types.

	The Total Number of Outpatient Visits = 1 (*n* = 3957)		The Total Number of Outpatient Visits = 2 (*n* = 1706)		The Total Number of Outpatient Visits ≥ 3 (*n* = 1881)		Overall (*n* = 7544)	
	Hospital Outpatient Visits (*n* = 1754)	Primary Care Outpatient Visits (*n* = 2203)	Hospital Outpatient Visits (*n* = 573)	Primary Care Outpatient Visits (*n* = 1133)	Hospital Outpatient Visits (*n* = 470)	Primary Care Outpatient Visits (*n* = 1411)	Hospital Outpatient Visits (*n* = 2797)	Primary Care Outpatient Visits (*n* = 4747)
ER visits, *n* (%)	137 *** (7.81)	24 (1.09)	31 *** (5.41)	12 (1.06)	24 *** (5.11)	14 (0.99)	192 *** (6.86)	50 (1.05)
Hospitalization, *n* (%)	463 *** (26.40)	325 (14.75)	148 *** (25.83)	187 (16.50)	142*** (30.21)	270 (19.14)	753 *** (26.92)	782 (16.47)
Number of hospitalizations, mean (SD)	0.41 *** (0.89)	0.23 (0.68)	0.47 *** (0.97)	0.27 (0.76)	0.56 *** (1.16)	0.32 (0.86)	0.45 *** (0.96)	0.27 (0.76)
Length of hospitalization in days, mean (SD)	3.51 *** (8.96)	1.52 (5.06)	3.16 *** (7.20)	1.70 (6.50)	4.11 *** (9.61)	2.01 (6.34)	3.54 *** (8.75)	1.73 (5.83)

*** *p* < 0.001. There were significant differences based on Chi-square test and Mann–Whitney U test in ER and hospital inpatient services utilization across different outpatient visit types in three subgroups and total sample.

**Table 4 ijerph-19-12979-t004:** Association between the outpatient visit types and ER or hospital inpatient services utilization among middle-aged and elderly Chinese participants.

Outpatient Visit Types	ER Visits or Not *n* = 7544		Hospitalization or Not *n* = 7544		Number of Hospitalizations *n* = 1535		Length of Hospitalization Days *n* = 1535	
	Adjusted OR (95% CI)	Average Marginal Effect (95% CI)	Adjusted OR (95% CI)	Average Marginal Effect (95% CI)	Adjusted IRR (95% CI)	Average Marginal Effect (95% CI)	Adjusted IRR (95% CI)	Average Marginal Effect (95% CI)
Ref: Hospital outpatient visits								
Primary care outpatient visits	0.141 *** (0.101, 0.194)	−0.058 *** (−0.069, −0.048)	0.623 *** (0.546, 0.711)	-0.069 *** (−0.089, −0.049)	0.961 (0.788,1.173)	−0.020 (−0.122, 0.081)	0.886 ** (0.810, 0.969)	−1.407 ** (−2.450, −0.365)

** *p* < 0.01, *** *p* < 0.001. All models were adjusted for economic region, PGDP, current residence, self-reported health status, ADL limitations, IADL limitations, depression, age group, sex, marital status, education, work status, smoking, drinking, number of chronic diseases, frequency of social activity, medical insurance status, PCE, number of caregivers, and living arrangement.

**Table 5 ijerph-19-12979-t005:** Trend test of association between the outpatient visit types and ER or hospital inpatient services utilization among middle-aged and elderly Chinese participants.

Outpatient Visit Types	ER Visits or Not ^a^ *n* = 7544		Hospitalization or Not *n* = 7544		Number of Hospitalizations ^b^ *n* = 1535		Length of Hospitalization Days *n* = 1535	
	Adjusted OR (95% CI)	Average Marginal Effect (95% CI)	Adjusted OR (95% CI)	Average Marginal Effect (95% CI)	Adjusted IRR (95% CI)	Average Marginal Effect (95% CI)	Adjusted IRR (95% CI)	Average Marginal Effect (95% CI)
The total number of outpatient visits = 1	*n* = 3957				*n* = 788			
Ref: Hospital outpatient visits								
Primary care outpatient visits	0.131 *** (0.084, 0.204)	−0.065 *** (−0.079, −0.052)	0.577 *** (0.481, 0.692)	−0.076 *** (−0.101, −0.051)	1.096 (0.871, 1.380)	0.086 (−0.132, 0.305)	0.828 ** (0.740, 0.927)	−2.199 ** (−3.503, −0.896)
The total number of outpatient visits = 2	*n* = 1706				*n* = 335			
Ref: Hospital outpatient visits								
Primary care outpatient visits	0.154 *** (0.073, 0.327)	−0.048 *** (−0.070, −0.026)	0.724 * (0.547, 0.958)	−0.046 *** (−0.086, −0.005)	0.821 (0.591, 1.140)	−0.200 (−0.532, 0.132)	0.967 (0.806, 1.160)	−0.371 (−2.353, 1.611)
The total number of outpatient visits ≥ 3	*n* = 1881				*n* = 412			
Ref: Hospital outpatient visits								
Primary care outpatient visits	0.173 *** (0.083, 0.359)	−0.042 *** (−0.063, −0.020)	0.684 * (0.506, 0.923)	−0.059 *** (−0.107, −0.011)	0.733 (0.511, 1.053)	−0.263 (−0.595, 0.069)	0.974 (0.834, 1.137)	−0.312 (−2.140, 1.515)

* *p* < 0.05, ** *p* < 0.01, *** *p* < 0.001. ^a^ There was a linear trend of a significant increase in the odds of ER visits as the total number of outpatient visits increased by trend test. ^b^ There was a linear trend of a significant decrease in the number of hospitalizations as the total number of outpatient visits increased by trend test. All models were adjusted for economic region, PGDP, current residence, self-reported health status, ADL limitations, IADL limitations, depression, age group, sex, marital status, education, work status, smoking, drinking, number of chronic diseases, frequency of social activity, medical insurance status, PCE, number of caregivers, and living arrangement.

## Data Availability

Not applicable.

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
