# Peer review of "Association between Primary Care Utilization and Emergency Room or Hospital Inpatient Services Utilization among the Middle-Aged and Elderly in a Self-Referral System: Evidence from the China Health and Retirement Longitudinal Study 2011–2018"

_ijerph, 2022, doi:10.3390/ijerph191912979_

Round 1

Reviewer 1 Report

General comments:

This study used several statistical methods to examine differences in the utilization of emergency room (ER) and hospital inpatient services between middle-aged and elderly individuals using hospital outpatient services and primary care outpatient services. The topic is meaningful but the study design has some defects. Although the study has controlled for some potential confounders, such as age, gender, and health status, the selection bias remained. Those treated in primary facilities generally have relatively mild disease. Therefore, they were less likely to have ER visits or hospitalization. The results therefore may be contrary to the actual association. Moreover, the language should be polished better and the structure of the article needs to be readjusted.

major revision

1.      Although the study has controlled for some potential confounders, such as age, gender, and health status, the selection bias remained. Those treated in primary facilities generally have relatively mild disease. Therefore, they were less likely to have ER visits or hospitalization.

2.      The language should be polished better and the format of the table needs to be standardized.

3.      The structure of the article needs to be readjusted. For example, the introduction of the China Health and Retirement Longitudinal Study (CHARLS) should be presented in the Methodology section. The background in the Methods section should be presented in the Introduction section.

4.      The conclusions and discussions should correspond to the results of the study.

Author Response

Response to Reviewer 1 Comments

Point 1: Although the study has controlled for some potential confounders, such as age, gender, and health status, the selection bias remained. Those treated in primary facilities generally have relatively mild disease. Therefore, they were less likely to have ER visits or hospitalization.

Response 1: First of all, the authors would like to thank this reviewer for the valuable suggestion.  In terms of the comment, we performed extra analyses and made the following adjustments.

First, we moved the Appendix B: descriptive statistics of covariates, 2011-2018 to the “Results” section in the main text, and split the original Table A2 in Appendix B into Table 1 and Table 2 by waves, 2011-2018 (Line 320 Line 328, respectively). As shown in Table 2, there was no statistically significant differences across self-reported health between primary care outpatient visits respondents and hospital outpatient visits respondents. This, somehow, indicates that the selection bias was not in this cohort since the usage of primary care (versus hospital) outpatients is not associated with patients’ health status (Although it is self-reported health status, it still can be a good proxy for patients’ true health status).  We have added content to the Descriptive Statistics subsection with the Results section (Line 315- 319).

Second, we added a stratified analysis by respondents’ self-reported health status and found consistent results, and found that the association between exposure (PC outpatient usage) and the outcome (inpatient usage) did not change by different health status. This is showed that the association between the exposure and outcome did not modify by the health status (A model with an interaction will do this too. But since the effect modification is not the main focus in this work, we just made it ad hoc). We briefly mentioned the stratified analysis results in the “Discussion” section (Line 444-450) and have included the stratified analysis results in Appendix B (Line 580).

Finally, we have explained in both the “Introduction” section (Line 106-108) and the “Discussion” section (Line 444-450) that the ER in China do not only admit critically illed patients.  Patients with less several conditions (such as throat infections) may use hospitals’ ER and can usually afford it and/or get covered by their insurance. This somehow weaken the association between ER usage and the severity of their disease. We hope that the adjusted will meet with approval. The specific contents are as follows:

Line 106-108:

In addition, patients may choose the ER as their first point of contact. It does not turn away patients even for minor health problems in most cases [20, 21].

Line 315- 319:

More importantly, there was no difference in self-reported health status between primary care and hospital outpatient visits respondents, which somehow implies that respondents' use of different types of outpatient services (primary care vs. hospital) was not associated with respondents’ health status.

Line 444-450:

Moreover, we also conducted a stratified analysis by respondents’ self-reported health status and found consistent results (Appendix B). This is showed that the association between the primary care usage and ER or inpatient usage did not modify by the health status. Moreover, in self-referral system in China, ER do not only treat critically ill patients, patients with minor health problems may use ER. This somehow weaken the association between ER usage and the severity of the patient's disease.

Point 2: The language should be polished better and the format of the table needs to be standardized.

Response 2: Thank you for your suggestion. We have tried our best to polish the language in the revised manuscript, and we also invited a native English speaker from the USA to help polish our article. We have also modified the table formatting to make it more pleasing and standard. We hope the languages in this revised manuscript is acceptable.

Point 3: The structure of the article needs to be readjusted. For example, the introduction of the China Health and Retirement Longitudinal Study (CHARLS) should be presented in the Methodology section. The background in the Methods section should be presented in the Introduction section.

Response 3: Thank you for your suggestion. We have removed the description of CHARLS in the “Introduction” section (Line103-115) and added a description of how CHARLS was performed in the “Data Sources” subsection within the “Methods” section (Line 179-209). The specific contents are as follows:

Line 179-209:

The national baseline survey of CHARLS was conducted between June 2011 and March 2012, with subsequent follow-up every 2 years, and a total of four waves currently being updated. CHARLS adopted a stratified, multi-stage probability-proportionate-to-size (PPS) sampling strategy. In the first stage, all counties/districts in China (except for Tibet) were stratified by region, within region by rural counties or urban districts, and by per capita GDP. Subsequently, 150 counties/districts were randomly selected by PPS sampling. In the next stage, in each county/district, three villages/communities were randomly chosen using PPS sampling. Thus, this study was conducted in 28 provinces, 150 countries/districts, and 450 villages/urban communities across China. In the third stage, a dedicated mapping software (CHARLS-GIS) designed and developed by CHARLS was used to conduct field mapping and produce a list of dwelling units in each village/community, from which a number of dwellings were then randomly selected. If there was more than one household meeting the age eligibility in a given dwelling, one household was randomly selected. Finally, in each sampled household, a short screening form from household questionnaire was used to screen out whether a member meet the age eligibility in the household. If there were members aged >45 years in a household, one member was randomly selected; a selected member aged >45 years was automatically chosen as the main respondent [31]. The core household questionnaire includes the following sections: demographics; family structure/transfer; health; health insurance and health care utilization; work, retirement and pension; Income, expenditures and assets; housing characteristics and the community and policy modules. To ensure the standardization of the interviews and the accuracy of the data, CHARLS uniformly provided rigorous training to the recruited university students and sent them to conduct field surveys across the country. During the fieldwork, each respondent who agreed to participate in the survey had to sign two informed consent forms.

Point 4: The conclusions and discussions should correspond to the results of the study.

Response 4: We added secondary headings to the “Results”  section (Line 301-302, Line 355-356, Line 375-376) and “Discussion” section (Line 416-417, Line 439-440, Line 506-507 and Line 535) so that the “Results”, “Discussion” and “Conclusion” correspond to the two research questions. We hope that this revision and layout will make the structure of the article clearer and thus more reader friendly. The specific contents are as follows:

Line 301:          3. Results

Line 302:          3.1. Descriptive Statistics

Line 355-356:  3.2. Association Between Outpatient Visits Types and ER or Hospital Inpatient Services Utilization

Line 375-376:  3.3. Trend Test of Association Between Outpatient Visits Types and ER or Hospital Inpatient Services Utilization

Line 416:           4. Discussion

Line 417:           4.1. Main Findings

Line 439-440:   4.2. Association Between Outpatient Visits Types and ER or Hospital Inpatient Services Utilization

Line 506-507:   4.3. Trend Test of Association Between Outpatient Visits Types and ER or Hospital Inpatient Services Utilization

Line 536:          4.4. Study Limitations

Reviewer 2 Report

I’ve read with attention this manuscript. This manuscript tries to compared the utilization of emergency room and hospital inpatient services among patients that used hospital outpatient services or primary care outpatient services. Overall, this is a clear, concise manuscript. The manuscript is well written, especially methods and results. However, I have some minor revision.

 The title is too long. In addition, abbreviations should not be used in the title.

Abstract:

You have said that the 2009 reforms increased the use of inpatient and emergency services. But primary care reduces the use of these services. Wasn't primary care part of the 2009 reforms?

The statement of the purpose of the study is vague for the reader. “This study aims to examine differences in the utilization of emergency room (ER) and hospital inpatient services between middle-aged and elderly individuals using hospital outpatient services and primary care outpatient services in a self-referral system.”

Introduction:

The references are very old. References 1 to 10 that form the introduction configuration should be updated.

You said that “Total health expenditures increased from ¥ 1.998 trillion in 2010 to ¥ 7.559 trillion in 2021”. However, the reference for this sentence was related to the 2019.

Methods:

The methodology is well written, especially the Assessment Framework section, which provides good information to the reader.

It is suggested to explain about the classification of health centers into primary care and hospital in China.

Discussion:

As shown in Figure 2, from 2011 to 2018, the rate of primary care outpatient visits decreased, whereas those using hospital outpatient services increased.

You should discuss these results in the discussion.

Also, the discussion of the article should be strengthened with new and updated references

Author Response

Response to Reviewer 2 Comments

Point 1: The title is too long. In addition, abbreviations should not be used in the title.

Response 1: Thank you for your suggestion. We have removed abbreviation of “ER” in the title and replace it with “emergency room”, and we have added a more concise running title of our work. The specific contents are as follows:

Running title: Primary care utilization and inpatient services usage.

Title: Association between primary care utilization and emergency room or hospital inpatient services utilization among the middle-aged and elderly in a self-referral system: Evidence from the China Health and Retirement Longitudinal Study 2011–2018.

Point 2: Abstract: You have said that the 2009 reforms increased the use of inpatient and emergency services. But primary care reduces the use of these services. Wasn't primary care part of the 2009 reforms?

Response 2: Thank you for your valuable comment and we apologize for this error in the Abstract. We have revised this sentence according to your comment. The specific contents are as follows:

Line 20-24 :

With the rapid economic growth and aging, hospital inpatient and emergency services utilization has grown rapidly, and put forward an urgent requirement to adjust and optimize the structure of health service utilization. Studies have shown that primary care is an effective way to reduce inpatient and emergency service utilization.

Point 3: Abstract: The statement of the purpose of the study is vague for the reader. “This study aims to examine differences in the utilization of emergency room (ER) and hospital inpatient services between middle-aged and elderly individuals using hospital outpatient services and primary care outpatient services in a self-referral system.”

Response 3: Thank you for your suggestion. We have revised this sentence to make it clearer. The specific contents are as follows:

Line 24-27:

This study aims to examine whether middle-aged and elderly individuals who selected primary care outpatient service in the last month had less emergency room (ER) and hospital inpatient service utilization than those who selected hospitals outpatient service via the self-referral system.

Point 4: Introduction: The references are very old. References 1 to 10 that form the introduction configuration should be updated.

Response 4: We have updated the references in the “Introduction” section as much as possible. We have removed unnecessary and older references and replaced it with newer references. The specific updates are as follows:

Line 590-602:

  1. WHO. Caring for the health of the elderly in China. Available online: https://www.who.int/news-room/feature-stories/detail/caring-for-the-health-of-the-elderly-in-china (accessed on 28 May 2021).
  2. Starfield, B.; Shi, L.; Macinko, J. Contribution of primary care to health systems and health. Milbank Q 2005, 83, 457-502.
  3. Ellner, A. L.; Phillips, R. S. The Coming Primary Care Revolution. J Gen Intern Med 2017, 32, 380-386.
  4. van Gool, K.; Mu, C.; Hall, J. Does more investment in primary care improve health system performance? Health Policy 2021, 125, 717-724.
  5. WHO. Primary health care measurement framework and indicators: monitoring health systems through a primary health care lens; World Health Organization: Geneva, Switzerland, 2022.
  6. National Health Commission. China Health Statistics Yearbook 2010. Available online: http://www.nhc.gov.cn/zwgk/tjnj1/ejlist.shtml (accessed on 8 October 2010).
  7. National Health Commission. China Health Statistics Yearbook 2021. Available online: http://www.nhc.gov.cn/ (accessed on 8 October 2021).

Point 5: Introduction: You said that “Total health expenditures increased from ¥ 1.998 trillion in 2010 to ¥ 7.559 trillion in 2021”. However, the reference for this sentence was related to the 2019.

Response 5: We apologize for the misunderstanding caused by not citing the reference in this sentence. We have added new references for this sentence. The specific references are as follows:

Line 97-98:

Total health expenditures increased from ¥ 1.998 trillion in 2010[16]to ¥ 7.559 trillion in 2021[15]

Line 621-626:

  1. National Health Commission. Statistical Bulletin on the Development of Healthcare in China. 2021. Available online: http://www.nhc.gov.cn/guihuaxxs/s3586s/202207/51b55216c2154332a660157abf28b09d.shtml (accessed on 12 July 2022).
  2. National Health Commission. China Health Statistics Yearbook 2011. Available online: http://www.nhc.gov.cn/mohwsbwstjxxzx/s7967/201301/f34860a7bdb64aad9029eb860a6cf4d2.shtml (accessed on 16 January 2013).

Point 6: Methods: It is suggested to explain about the classification of health centers into primary care and hospital in China.

Response 6: We have added the classification of health centers in China in the “2.1 Assessment Framework” subsection within the “Method” section. The specific contents are as follows:

Line 142-147:

According to the National Center for Health Statistics in China, healthcare providers in the primary care system include community healthcare centers, township hospitals, healthcare posts, and village clinics/private clinics. In the hospital care system, healthcare providers include general hospitals, specialized hospitals, and Chinese medicine hospitals.

Point 7: Discussion: As shown in Figure 2, from 2011 to 2018, the rate of primary care outpatient visits decreased, whereas those using hospital outpatient services increased. You should discuss these results in the discussion.

Response 7: We have added a discussion point of the result in the “Discussion” section. The specific contents are as follows:

Line 422-430:

This study identifies a decreasing temporal trend in primary care out-patient visits rates, and an increasing trend in hospital outpatient visits rates in China from 2011 to 2018. It suggests that, during this period of 2011 to 2018, patients in China tended to depends more and more on large hospitals through the self-referral system, which showed that China’s healthcare system is rapidly moving towards a hospital-oriented mode. This standing trend further warranted our exploration of the usage of primary care in China’s healthcare system.

Point 8: Discussion: Also, the discussion of the article should be strengthened with new and updated references.

Response 8: Thanks for your kind reminder. We have updated the references in the “Discussion” section as much as possible. We have removed unnecessary older references and replaced it with newer ones. The specific references are as follows:

  1. Liao, R.; Liu, Y.; Peng, S.; Feng, X. L. Factors affecting health care users' first contact with primary health care facilities in north eastern China, 2008-2018. BMJ Glob Health 2021, 6.
  2. Riedl, B.; Kehrer, S.; Werner, C. U.; Schneider, A.; Linde, K. Do general practice patients with and without appointment differ? Cross-sectional study. BMC Fam Pract 2018, 19, 101.
  3. Salisbury, C.; Munro, J. Walk-in centres in primary care: a review of the international literature. Br J Gen Pract 2003, 53, 53-9.
  4. Engstrom, S.; Foldevi, M.; Borgquist, L. Is general practice effective? A systematic literature review. Scand J Prim Health Care 2001, 19, 131-44.
  5. Kim, S. L.; Tarn, D. M. Effect of Primary Care Involvement on End-of-Life Care Outcomes: A Systematic Review. J Am Geriatr Soc 2016, 64, 1968-1974.
  6. Pereira Gray, D. J.; Sidaway-Lee, K.; White, E.; Thorne, A.; Evans, P. H. Continuity of care with doctors-a matter of life and death? A systematic review of continuity of care and mortality. BMJ Open 2018, 8, e021161.
  7. Kao, Y. H.; Lin, W. T.; Chen, W. H.; Wu, S. C.; Tseng, T. S. Continuity of outpatient care and avoidable hospitalization: a systematic review. Am J Manag Care 2019, 25, e126-e134.
  8. Godard-Sebillotte, C.; Strumpf, E.; Sourial, N.; Rochette, L.; Pelletier, E.; Vedel, I. Primary care continuity and potentially avoidable hospitalization in persons with dementia. J Am Geriatr Soc 2021, 69, 1208-1220.
  9. Dyer, S. M.; Suen, J.; Williams, H.; Inacio, M. C.; Harvey, G.; Roder, D.; Wesselingh, S.; Kellie, A.; Crotty, M.; Caughey, G. E. Impact of relational continuity of primary care in aged care: a systematic review. BMC Geriatr 2022, 22, 579.

Reviewer 3 Report

Dear Authors, 

It is a detailed manuscript well-presented one, giving extensive insight on the Health care system of China and its flaws and possibilities to improve. Authors have made efforts to identify the issues which require attention. But at the same time, it becomes important to be specific on how the surveys were performed. No details are added to this regard. Copy of questionnaires used, whether verbal or written, was it at the time of their visit to the family physician/hospital or a home-to-home survey data collection. How were the surveys coordinated, keeping in mind the population of China. Methodology lacks a lot of such clear specifications. Especially in regard to the waves mentioned and CHARLS. Explanation in this regard, will clarify the readers and add value to the article. Authors can also compare the changes seen during the pandemic in such hospital/outpatient visits. 

Author Response

Response to Reviewer 3 Comments

Point 1: It is a detailed manuscript well-presented one, giving extensive insight on the Health care system of China and its flaws and possibilities to improve. Authors have made efforts to identify the issues which require attention. But at the same time, it becomes important to be specific on how the surveys were performed. No details are added to this regard. Copy of questionnaires used, whether verbal or written, was it at the time of their visit to the family physician/hospital or a home-to-home survey data collection. How were the surveys coordinated, keeping in mind the population of China. Methodology lacks a lot of such clear specifications. Especially in regard to the waves mentioned and CHARLS. Explanation in this regard, will clarify the readers and add value to the article. Authors can also compare the changes seen during the pandemic in such hospital/outpatient visits.

Response 1: Thank you very much for your rigorous attitude and professional advice. We have added a detailed description of how CHALRS was performed in the “Data Sources” subsection within the “Methods” section. In addition, the data we used CHARLS 2011-2018 is the most recent data that has been publicly released so far, the data during the COVID-19 pandemic is not available, so it is not possible to compare changes in outpatient visits during the pandemic. Thank you for your suggestion, this may be a future research point. The specific contents are as follows:

Line 179-209:

The national baseline survey of CHARLS was conducted between June 2011 and March 2012, with subsequent follow-up every 2 years, and a total of four waves currently being updated. CHARLS adopted a stratified, multi-stage probability-proportionate-to-size (PPS) sampling strategy. In the first stage, all counties/districts in China (except for Tibet) were stratified by region, within region by rural counties or urban districts, and by per capita GDP. Subsequently, 150 counties/districts were randomly selected by PPS sampling. In the next stage, in each county/district, three villages/communities were randomly chosen using PPS sampling. Thus, this study was conducted in 28 provinces, 150 countries/districts, and 450 villages/urban communities across China. In the third stage, a dedicated mapping software (CHARLS-GIS) designed and developed by CHARLS was used to conduct field mapping and produce a list of dwelling units in each village/community, from which a number of dwellings were then randomly selected. If there was more than one household meeting the age eligibility in a given dwelling, one household was randomly selected. Finally, in each sampled household, a short screening form from household questionnaire was used to screen out whether a member meet the age eligibility in the household. If there were members aged >45 years in a household, one member was randomly selected; a selected member aged >45 years was automatically chosen as the main respondent[31]. The core household questionnaire includes the following sections: demographics; family structure/transfer; health; health insurance and health care utilization; work, retirement and pension; Income, expenditures and assets; housing characteristics and the community and policy modules. To ensure the standardization of the interviews and the accuracy of the data, CHARLS uniformly provided rigorous training to the recruited university students and sent them to conduct field surveys across the country. During the fieldwork, each respondent who agreed to participate in the survey had to sign two informed consent forms.

Round 2

Reviewer 1 Report

The authors revised the manuscript addressing all the issues as required and it's suitabe for publication